# Effect of Temperature and Ionic Substitutions on the Tegumental Potentials of Protoscoleces of *Echinococcus granulosus*

**DOI:** 10.3390/tropicalmed8060303

**Published:** 2023-06-02

**Authors:** Mónica Patricia Antonella Carabajal, María José Fernández Salom, Santiago Olivera, Horacio F. Cantiello

**Affiliations:** Laboratorio de Canales Iónicos, Instituto Multidisciplinario de Salud, Tecnología y Desarrollo (IMSaTeD, CONICET-UNSE), Santiago del Estero 4206, Argentina; carabajalantonella@gmail.com (M.P.A.C.); mjfsalom2439@gmail.com (M.J.F.S.); santiagosolivera@hotmail.com (S.O.)

**Keywords:** *E. granulosus*, tegumental potential, active transport, electrodiffusional ion movement

## Abstract

The protoscolex (PSC) is generated by asexual reproduction at the larval stage of taeniid *Echinococcus granulosus* that causes cystic echinococcosis or hydatidosis, a worldwide zoonosis. The PSC is enveloped by a complex cellular syncytial tegument responsible for ionic movements and the hydroelectrolytic balance of the parasite. We recently reported on two electrical potentials in bovine lung protoscoleces (PSCs) that reflect differences in ionic movements between the parasite’s invaginated and evaginated developmental stages. Here, we explored the effect of temperature and ionic substitutions on the tegumental potentials of bovine lung PSCs of *Echinococcus granulosus* by microelectrode impalements. We observed that the transient peak potential was temperature-dependent, consistent with an active transport component in the invaginated state only. Further changes in the electrical potentials by high K^+^ depolarization, low external Ca^2+^, and addition of the diuretic amiloride are in agreement with the presence of a Ca^2+^-sensitive cation-selective electrodiffusional pathway in the outer surface of the parasite. Variations in electrical potential differences through the tegument provide an accessible and valuable parameter for studying ionic transport mechanisms and, therefore, potential targets for developing novel antiparasitic drugs.

## 1. Introduction

*Echinococcus granulosus* is a parasitic tapeworm in the class Cestoda in the phylum Platyhelminthes. Although it is a small parasite (only a few millimeters long), it is a typical taenia requiring two mammalian hosts to complete its life cycle of infecting animals and humans [1]. After hatching from the eggs, oncospheres develop into hydatid cysts that typically reside in the intermediate host’s liver and lungs encased in a cyst wall and containing hydatid fluid, causing hydatidosis or cystic echinococcosis (CE). The innermost germinal layer of the cyst wall gives rise to protoscoleces (PSCs) that later mature into adult worms when ingested by the definitive host [2].

CE is one of the most prevalent and neglected zoonotic diseases worldwide, according to data from the World Health Organization [3]. It is endemic in several South American countries, including Argentina, Chile, Peru, Uruguay, and southern Brazil [4,5]. In Argentina, the prevalence of CE is between 5.8% and 13.3%, with livers and lungs being the most frequently involved organs [6,7]. There is still much to be discovered regarding optimal chemotherapeutic treatments for this disease.

*E. granulosus* has no digestive system and all metabolic interchange occurs across the syncytial outer covering, the tegument. This external layer is a unique feature that is critical to its survival and successful parasitism. The tegument protects the parasite’s body from the host immune system and facilitates nutrient uptake, waste excretion, and host tissue invasion. Thus, it is essential to study and identify the molecular transport mechanisms to understand the physiology and, eventually, its applications in developing adequate chemotherapy [8].

Changes in the electrical potentials and a preliminary description of ion channel species have been observed in protoscoleces of *E. granulosus* from the ovine origin [9,10,11]. In addition, different genes encoding ion channels in *E. granulosus* have been identified [12,13]; however, no ion channel species has been yet characterized in the parasite.

The mode of action of the major groups of anthelmintic drugs (e.g., avermectins) is through selective ion channel regulation [14]. Electrophysiological techniques are necessarily part of the investigation to study the anthelmintic action of these drugs [15].

The tegumental electrical potentials reflect the ionic movements throughout the syncytial membranes, providing helpful information on the physiology of the parasite. Microelectrode recording is a useful electrophysiological technique that helps assess PSCs’ electrical properties and can help in the knowledge of electrolyte transport by the tegumental epithelium. This technique has been used in *Schistosoma mansoni* [16] and *E. granulosus* of ovine origin [17], where considerable changes have been observed upon immunological and chemical manipulations [18,19]. We recently used microelectrode recordings to gain insight into the electrical activity of bovine lung PSCs of *E. granulosus* in a standard saline solution [20]. We observed two distinct intraparasitic potentials (a transient peak potential and a stable second potential), most likely representing tegumental and intraparasitic extracellular space electrical potential differences, respectively. These values changed depending on the developmental status of the parasite, its anatomical regions, or time course after harvesting. 

Here, we applied the same electrophysiological technique to explore changes in electrical conductance elicited by ionic replacements, temperature, and the effect of an epithelial Na^+^-channel inhibitor, amiloride. 

## 2. Materials and Methods

### 2.1. Parasite Collection

PSCs of *E. granulosus* were obtained from pulmonary hydatid cysts of naturally infected cows slaughtered at a local abattoir. The parasites were collected, washed, and resuspended in PBS (pH 7.4) supplemented with penicillin (100 units/mL), streptomycin (100 µg/mL), and amphotericin (0.25 µg/mL). They were stored at 4 °C for six days after harvesting. The parasite’s viability was evaluated using the methylene blue exclusion method and microscope examination (×10) of body movements. 

### 2.2. Saline Solutions and In Vitro Incubation Procedures

PSCs were transferred and incubated in a Ringer–Krebs solution (RKS) containing 121 mM NaCl, 5 mM KCl, 22.5 mM MgCl_2_, 2.5 mM CaCl_2_, 10 mM HEPES, and 5.6 mM glucose, pH 7.4. Before each experiment, PSCs were washed and preincubated for 2 h at 37 °C or 4 °C in fresh RKS without antibiotics. In the case of ion substitution experiments, potassium concentrations were modified by raising KCl to 110 mM and reducing NaCl concentration to 15 mM to maintain Cl^−^ concentration. Low calcium conditions were reached with the addition of 1 mM EDTA. Amiloride hydrochloride hydrate (≥98%) was acquired from Sigma Aldrich Co (St. Louis, MO, USA). All other constituents, osmolality, and pH were preserved as the normal RKS.

### 2.3. Microelectrode Measurements

Electrical recordings were obtained, as recently detailed [20]. Briefly, we used a single microelectrode high input impedance (>10^11^ Ω) amplifier–intracellular electrometer (Model IE-210, Warner Instruments, Hamden, CT, USA) with an internal 4-pole low-pass Bessel filter set at 20 kHz and a sampling rate of 10 kHz. The electrometer was connected in parallel to an analog–digital converter (TL-1 interface. Tecmar, Solon, OH, USA) that fed the digital input of a personal computer running Axograph (Axon Instruments, Union City, CA, USA) as a digital oscilloscope. Microelectrodes from glass capillaries (Biocap, Buenos Aires, Argentina), with 1.25 mm internal diameter, were pulled on a PB-7 pipette puller and heat-polished on an MF-9 pipette polisher (Narishige, Tokyo, Japan) and filled with filtered 3 M KCl solution. The reference electrode was a wider-tip glass capillary, filled with 3 M KCl solution, connected to a Cl-plated silver wire (Ag/AgCl) and the ground socket of the electrometer. Tip resistance ranged between 10 and 40 MW. The recording chamber consisted of a glass slide, where an aliquot (500 μL) of a parasite suspension was added (~100 PSCs/mL). Parasites were impaled after being individually captured and held with a suction micropipette (Figure 1A). Impalements were performed under optical microscopy with an IMT2 Olympus inverted microscope (×10). 

### 2.4. Data Analyses and Statistics

The original tracings obtained from microelectrode impalements were analyzed with Clampfit 10.7 (Axon Instruments). Statistical analysis and data graphics were conducted with Sigmaplot 11.0 (Jandel Scientific, Corte Madera, CA, USA). Unless otherwise indicated, only parasite values up to 6 days post-harvest were considered for statistical analyses. The distribution of raw data that departed significantly from normality, as evaluated by the Shapiro–Wilk W test, was processed and statistically analyzed, as recently reported [20]. The nonparametric Mann–Whitney rank sum test was first applied, where box whisker plots were compared. The Box–Cox transformation also normalized the data to further conduct parametric statistics. The Student’s *t*-test and one-way ANOVA were used to determine statistical significance between experimental groups of transformed data. The significance level was established by Tukey’s post hoc test, regarding *p* values < 0.05 as statistically significant. In the present study, we expressed the average of corrected data values as the mean ± SEM for each experimental condition per number of individual PSCs impaled (*n*). 

## 3. Results

### 3.1. Effect of Low Temperature on Tegumental Membrane Potentials

Potential differences of invaginated and spontaneously evaginated PSCs were explored by microelectrode impalement, as recently reported [20]. Typical recordings obtained were reproduced in Figure 1B. Impalements rendered a rapid negative transient deflection in electrical potential, referred to as “peak” or potential difference 1 (PD_1_), that spontaneously decayed to a lower value, referred to as “plateau” potential (PD_2_). Relative to the bath (0 mV), mean values for invaginated PSCs were −70.8 ± 3.7 mV (*n* = 55) and −26.6 ± 0.5 mV (*n* = 55) for PD_1_ and PD_2_, respectively, representing a ∆*PD* (PD_1_ − PD_2_) of −44.2 mV (Table 1). Instead, evaginated parasites had a mean PD_1_ value of −93.6 ± 0.3 mV (*n* = 55) that decreased to a mean PD_2_ value of −34.0 ± 3.3 mV (*n* = 55), representing a ∆*PD* of −59.6 mV (Table 1). The mean PD_1_ and PD_2_ values and differences (∆*PD*) were statistically different between groups, suggesting other tegumental electric properties between the invaginated and evaginated states of the PSCs, being more negative in the evaginated state (*p* < 0.0001).

We measured trans-tegumental potentials to explore whether active transport may contribute to the tegumental potentials in PSCs pre-incubated at 4 °C. Enzymatically driven active transport is highly temperature dependent and reduced at low temperatures [21,22,23,24]. Under these conditions, the percentage population of evaginated PSCs decreased to 30%, compared with observations at 37 °C. Invaginated PSCs displayed a remarkably hyperpolarized PD_1_ of −140.0 ± 5.2 mV (*n* = 31) that turned into a PD_2_ of −39.1 ± 2.9 mV (*n* = 31), both values being significantly higher than those at 37 °C (*p* < 0.001). The ∆*PD*_1_ (PD_1_^37 °C^ − PD_1_^4 °C^) indicated a hyperpolarization of 69.2 mV, while the ∆*PD*_2_ (PD_2_^37 °C^ − PD_2_^4 °C^) was 12.5 mV (Figure 2A). Evaginated PSCs, however, showed mean values of −86.2 ± 9.2 mV (*n* = 15) and −25.9 ± 2.8 mV (*n* = 15) for PD_1_ and PD_2_, respectively (Figure 2B). The ∆*PD*_1_ indicated instead a depolarization of −7.4 mV, which was similar to ∆*PD*_2_, −8.1 mV (Figure 2A). The data agreed with an externally located active transport that depolarized the tegument in the invaginated state only. 

### 3.2. Effect of High External Potassium Concentration

To determine the voltage dependence and the contribution of external cations to the trans-tegumental potentials, we performed impalements in membrane-depolarized PSCs exposed to a high K^+^ solution (Na^+^ replacement). Membrane depolarization largely reduced the PD_1_ component under normal- and low-temperature conditions (Figure 2C,D). In experiments conducted at 37 °C, invaginated PSCs had mean values of −38.6 ± 4.1 mV (*n* = 30) and −21.2 ± 0.4 mV (*n* = 30) for the PD_1_ and PD_2_, respectively. This represented a 32.2 mV and 5.6 mV depolarization, respectively (Table 1). Thus, the ∆*PD* decreased from −44.2 mV to −17.6 mV between the normal (high Na^+^) and depolarized (high K^+^) conditions. On the other hand, evaginated PSCs had a PD_1_ of −64.1 ± 4.8 mV (*n* = 60) and PD_2_ of −26.7 ± 1.7 mV (*n* = 60) (Table 1). This represented a 29.5 mV and 7.3 mV depolarization, respectively. Thus, the ∆*PD* decreased from −59.6 mV to −37.4 mV between the normal (high Na^+^) and depolarized (high K^+^) conditions. Under these conditions, while PD_1_ was statistically lower under high external Na^+^ conditions, PD_2_ remained similar in both invaginated and evaginated PSCs (*p* > 0.05), indicating that the external tegument but not the internal compartment responded to high K^+^ depolarization.

### 3.3. Effect of Low External Calcium

To further explore the external ionic contributions to the trans-tegumental potentials of the PSCs, we also tested the effect of lowering external Ca^2+^ by adding the Ca^2+^ chelating agent EGTA (1 mM) to the bathing solution (Figure 3). The omission of external Ca^2+^ induces ionic permeability and morphological changes in epithelia [25,26]. At 37 °C, mean values for invaginated PSCs were −58.9 ± 4.4 mV (*n* = 7) and −18.0 ± 2.5 mV (*n* = 7) for PD_1_ and PD_2_, respectively. This represented a 12 mV and 8.6 mV depolarization to control, respectively. Thus, the ∆*PD* in low Ca^2+^ was −40.9 mV, statistically similar to the control value in the normal saline solution at 37 °C (44.2 mV). Additionally, evaginated PSCs showed marked differences from their respective controls, with mean values of −58.3 ± 7.5 mV (*n* = 26) and −19.5 ± 0.5 mV (*n* = 26) for PD_1_ and PD_2_, respectively. This meant a 35.3 mV depolarization for ∆*PD*_1_ and 14.5 mV for ∆*PD*_2_. The ∆*PD* (PD_1_ − PD_2_) for these PSCs was −38.8 mV, statistically lower than in the presence of normal Ca^2+^ (−59.6 mV), representing changes in both potentials.

### 3.4. Effect of Amiloride

We also tested the known diuretic and epithelial channel blocker amiloride for a possible effect of cation-permeable pathways on the spontaneous potentials of PSCs of *E. granulosus*. Pyrazinoylguanidine amiloride is a K^+^-sparing diuretic with a relatively high affinity for the epithelial Na^+^ ENaC channel (ENaC) and a lower affinity for several nonselective cation channels and other ion transporters [27,28]. In RKS, at 37 °C in the presence of amiloride (2 mM), invaginated PSCs showed a PD_1_ of −54.9 ± 5.7 mV (*n* = 4) and a PD_2_ of −19.6 ± 3.8 mV (*n* = 4). Both values were lower than their respective controls (15.9 mV and 7.0 mV, respectively). Differences were also observed in evaginated PSCs (13.5 mV and 15.8 mV, respectively), with mean values for PD_1_ and PD_2_ in amiloride of −80.1 ± 15.9 mV (*n* = 11) and −18.2 ± 3.0 mV (*n* = 11), respectively. In this case, PD_2_ was statistically different (*p* < 0.001) with respect to the control condition. Thus, the amiloride-sensitive ionic transport component affected parasitic potentials of the same magnitude only in the evaginated state. 

The effect was also tested in PSCs depolarized with high external K^+^ (data not shown). Under these conditions, obtaining or measuring invaginated PSCs was impossible. However, in evaginated PSCs incubated with amiloride, PD_1_ was −66.3 ± 8.7 mV (*n* = 30) and PD_2_ was −18.6 ± 0.5 mV (*n* = 30). These values represented nonstatistical differences of 2.2 mV and −8.1 mV changes for PD_1_ and PD_2_ with respect to the control, without amiloride at 37 °C (Table 1). Interestingly, the ∆*PD* (PD_1_ − PD_2_) showed a hyperpolarization of 10.3 mV, which largely depended on the changes in PD*_2_*. Finally, we also tested the effect under low external Ca^2+^ conditions in PSCs incubated at 37 °C in RKS with 1 mM EGTA (Figure 4). Invaginated PSCs had mean values of −35.1 ± 5.5 mV (*n* = 6) and −13 ± 2.7 mV (*n* = 6) for PD_1_ and PD_2_, respectively (Figure 4B). This represented −23.8 mV and −5 mV depolarizations with respect to their controls (no amiloride) in low Ca^2+^ (Figure 3 and Figure 4B). Interestingly, evaginated PSCs had mean values of −60.3 ± 0.3 mV (*n* = 20) and −19.2 ± 0.3 mV (*n* = 20) for the PD_1_ and PD_2_, respectively, which meant a hyperpolarization of only 2 mV and 6 mV, respectively (Figure 4B).

## 4. Discussion

The electrophysiology of worms has been extensively studied, particularly nematodes [29], while that of plathelminths has been dedicated to trematodes such as *Schistosoma mansoni*, where organismal potentials have been thoroughly explored [16,30,31,32]. Specific ion channel species targeting potentially relevant pharmaceutical species have recently been identified [33]. Current interest is identifying and exploring ion channels in other parasitic species, such as *Trypanosoma cruzi* [34]. 

This present study explores the electrical mechanisms associated with ionic movements of the larval stage of the cestode *Echinococcus granulosus,* with a methodology recently reported for cow lung PSCs [20]. In that study, we observed that, after the microelectrode’s entrance on the PSC’s tegumental surface, bimodal transient potentials appeared; the first, as a peak of negative potential (PD_1_), was lower in invaginated PSCs compared with evaginated PSCs. The peak potential always decayed spontaneously to a lower plateau value (PD_2_), lower in the invaginated state. We concluded that the first peak voltage deflection (PD_1_) would correspond to the trans-tegumental potential, as it was previously reported for ovine PSCs of *E. granulosus* [17] and *Schistosoma mansoni* [16,32]. The second potential, PD_2_, was attributed to muscle areas below the tegumental membrane, in agreement with electrophysiological studies describing the degree of electrical coupling between the tegument and muscle of *S. mansoni* [16,32,35]. 

Here, we used simple maneuvers, including changes in temperature and ionic substitutions, to identify potential ion transport mechanisms contributing to the electrical potentials of the PSCs.

Enzymatically driven active transport is highly temperature dependent and is eliminated at low temperatures [21,22,23,24]. To explore whether active transport contributed to the tegumental potentials, we conducted microelectrode impalements of the PSCs at 4 °C. The invaginated PSCs displayed remarkably hyperpolarized PD_1_ and PD_2_, significantly higher than those observed at 37 °C. However, the evaginated PSCs showed similar ∆*PD*_1_ and ∆*PD*_2_ values, in agreement with the presence of an externally located active transport that depolarized the tegument of the invaginated but not the evaginated PSCs.

To determine the voltage dependence and the contribution of external cations to the trans-tegumental potentials, we also performed impalements in membrane-depolarized PSCs exposed to a high K^+^ solution. The membrane depolarization reduced the PD_1_ component under normal- and low-temperature conditions. For the invaginated PSCs, the ∆*PD* decreased by 25.6 mV. In contrast, the ∆*PD* decreased by 22.2 mV for the evaginated PSCs, indicating that the external tegument but not the internal compartment responded to high K^+^ depolarization. Interestingly, the omission of external Ca^2+^, which induces ionic permeability and morphological changes in epithelia [25,26], decreased the ∆*PD* only in evaginated PSCs, which showed marked differences concerning their respective controls. 

Pyrazinoylguanidine amiloride is a K^+^-sparing diuretic with high affinity for the epithelial Na^+^ ENaC channel (ENaC) and lower affinity for several nonselective cation channels and other ion transporters [27,28]. Amiloride and its analogs, including 5-(N-ethyl-N-isopropyl) amiloride (EIPA), benzamil, and phenamil, have been extensively used as probes for a wide variety of transport systems [27]. Amiloride is a well-known antagonist of ENaC, Na^+^/Ca^2+^, Na^+^/H^+^ exchangers, nonselective cation channels, and voltage-gated K^+^ and Ca^2+^ channels [27,28,36,37,38,39,40,41]. Here, we tested amiloride for possible effect(s) on cation-permeable pathways to the spontaneous potentials of PSCs of *Echinococcus granulosus*. Under normal conditions (RKS at 37 °C), amiloride decreased PD_1_ and PD_2_ in both invaginated and evaginated PSCs. The amiloride-sensitive ionic transport component of both parasitic potentials was similar in the evaginated state. However, amiloride induced a 13.3 mV hyperpolarization, as observed by the differences in Δ*PD*. The effect of amiloride was also tested in evaginated PSCs depolarized with high external K^+^, which showed a 10 mV hyperpolarization largely dependent on PD_2_. Thus, the combined amiloride and high K^+^ maneuvers unmasked a nonselective electrodiffusional pathway, most consistent with a nonselective cation channel in the parasite under both developmental conditions. Future experiments will be required to identify the molecular nature of this transport mechanism. Low Ca^2+^-treated invaginated PSCs showed 24 mV and 5 mV depolarizations to their controls (no amiloride), while evaginated PSCs showed a hyperpolarization of only 2 mV and 6 mV, respectively.

The precise mechanisms underlying the generation of the electrical potential differences through ionic diffusion pathways in *Echinococcus* and other flatworms remain largely elusive. Cantiello et al. [42] noted that the passive K^+^ influx into ovine PSCs exhibited a simple diffusional mechanism, which was distributed across at least two distinct compartments with varying rates. However, no anatomical correlations were reported. It also preliminary demonstrated the presence of cation-selective channels in extracted membranes of *E. granulosus* PSCs from the ovine origin [10,11]. From the genome sequencing of the *E. granulosus* and other cestodes [13,43], it will be possible to identify different families of ion channels and transporters and to assess their potential relevancy as therapeutic targets [33]. There is no cure for cystic echinococcosis, with surgical treatment or chemotherapy as options [44]. The only drugs accepted for clinical treatment in humans are benzimidazole derivatives, including albendazole, mebendazole, and flubendazole, which interfere with microtubule polymerization [45]. One of the most effective antiparasitic drugs is praziquantel, which acts as a potential Ca^2+^ channel blocker [46,47]. New biological targets and relevant host–parasite interactions have the potential to introduce innovative therapeutic approaches [29,34,48]. Harnessing the emerging evidence, we have discovered a potential synergy between excitable cation channels and the actin cytoskeleton [49], which may augment the paralyzing effect of praziquantel on PSCs of *E. granulosus*, opening new avenues for therapeutic interventions [50].

To conclude, the present study showed that bovine lung PSCs of *Echinococcus granulosus* have an active transport mechanism only in the invaginated state. At the same time, the tegument of the PSC may have a cation-selective Ca^2+^-regulated and amiloride-sensitive electrodiffusional pathway in both developmental conditions. The microelectrode impalements used here to measure parasitic potentials may demonstrate to be a helpful, faster, and effective technique for assessing the involvement of different types of ion channels and enzymatic transporters in *E. granulosus*, enabling a rapid screening approach to investigate and uncover novel pharmacological targets with cestocidal action. Extending this evaluation to parasites found in other animals and humans holds promise in addressing crucial aspects of hydatid disease treatment and prevention.

## Figures and Tables

**Figure 1 tropicalmed-08-00303-f001:**
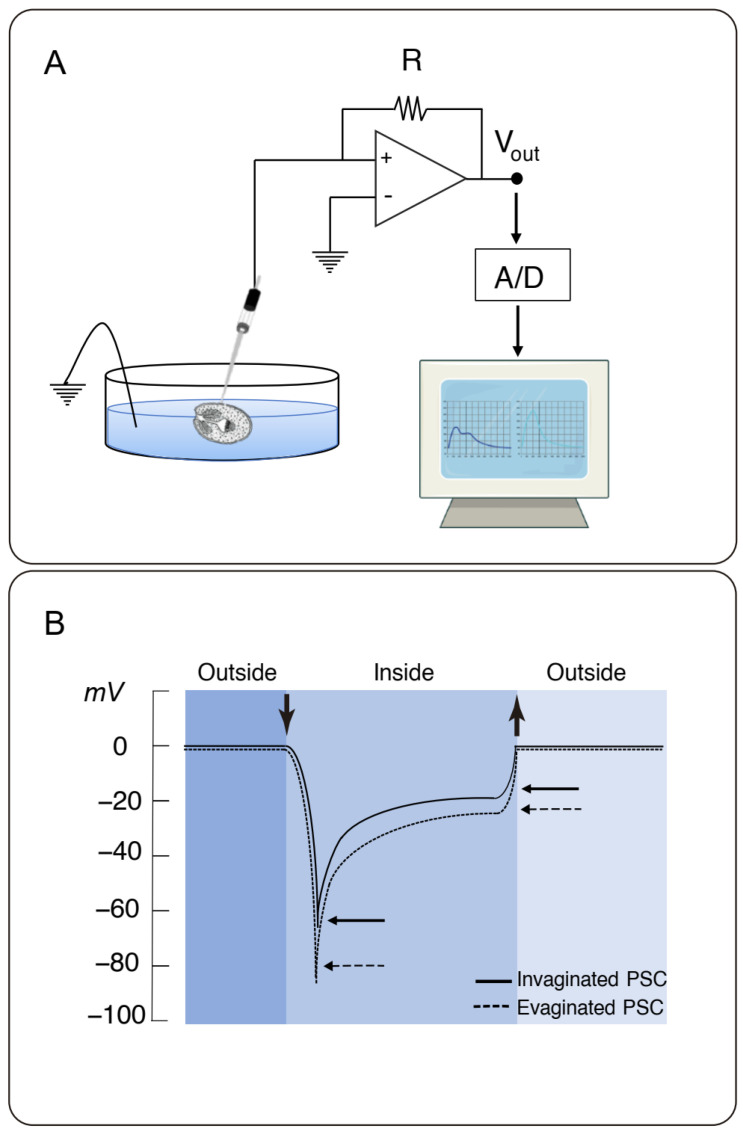
Experimental setup of the electrophysiological technique. (**A**) Schematics of the equipment used for electrical microelectrode recordings of PSCs from *E. granulosus*. Both ground and impaling microelectrodes were connected to an electrometer and recorded through an A/D system to a personal computer. (**B**) Typical recordings showing deflections upon impalement (downward vertical arrow) and withdrawal (upward vertical arrow). The tracings show an initial transient potential (peak, PD_1_) that decays to a lower potential (plateau, PD_2_). Horizontal arrows indicate recorded values for PD_1_ and PD_2_, respectively.

**Figure 2 tropicalmed-08-00303-f002:**
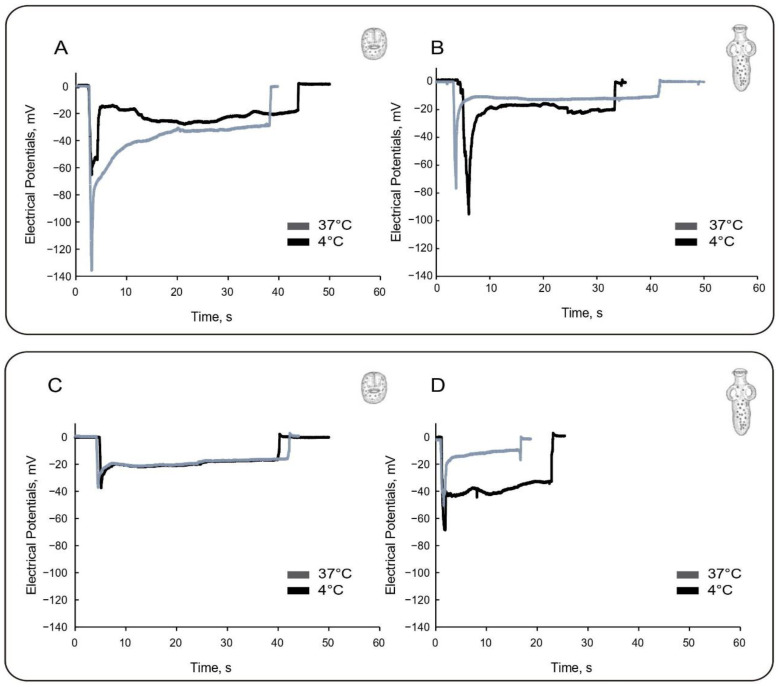
Effect of temperature and high potassium depolarization on the electrical potentials of *E. granulosus* PSC. (**A**,**B**) Time response of parasitic potentials from invaginated (**A**) and evaginated (**B**) PSCs in Ringer–Krebs solution (RKS). (**C**,**D**) Time response of parasitic potentials from invaginated (**C**) and evaginated (**D**) PSCs in high potassium concentration (110 mM KCl; modified RKS). These electrical recordings were obtained at either 37 °C (dark grey) or 4 °C (black).

**Figure 3 tropicalmed-08-00303-f003:**
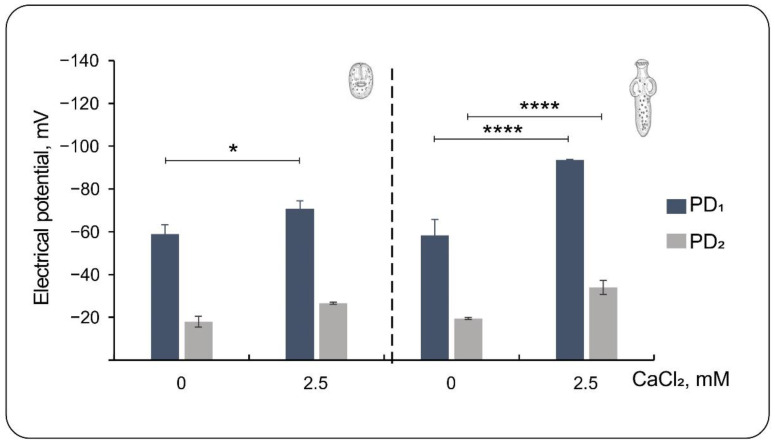
Effect of external Ca^2+^ reduction on the electrical potentials of PSC from *E. granulosus*. Values for invaginated PSCs (left) and evaginated PSCS (right) are expressed as changes in the normalized negative potentials (see Appendix A). PD_1_ represents the peak of the transient potential; PD_2_ represents the stable plateau potential. Values are the mean ± SEM for n between 30 and 60. Symbols * indicate statistically significant differences, with * *p* < 0.05 and **** *p* < 0.0001.

**Figure 4 tropicalmed-08-00303-f004:**
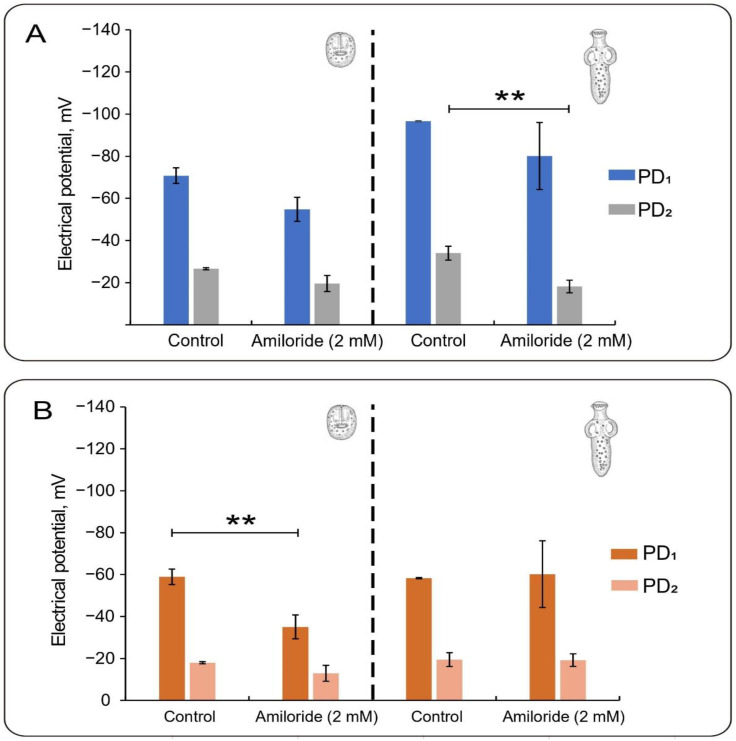
Effect of amiloride on the electrical potentials of PSC from *E. granulosus*. (**A**) PD_1_ and PD_2_ values for invaginated (left) and evaginated PSCs (right) in 2.5 mM Ca^2+^ (RKS) with or without amiloride. (**B**) PD_1_ and PD_2_ values for invaginated (left) and evaginated PSCs (right) in low external calcium (1 mM EGTA) with or without amiloride. Electrical potentials are expressed as changes in the normalized negative potentials (see Appendix A). Values are the mean ± SEM for n between 30 and 60. ** indicate the statistically significant difference with *p* < 0.01.

**Table 1 tropicalmed-08-00303-t001:** Effect of temperature and ionic substitutions on the tegumental potentials of protoscoleces of *E. granulosus*.

Ionic Composition (mM)	T (°C)	Invaginated PSC		Evaginated PSC	
		PD_1_	PD_2_	PD_1_	PD_2_
121 Na^+^ 5 K^+^ 2.5 Ca^2+^	37	−70.8 ± 3.7 (55)	−26.6 ± 0.5 (55)	−93.6 ± 0.3 (55)	−34.0 ± 3.3 (55)
	4	−140.0 ± 5.2 (31)	−39.1 ± 2.9 (31)	−86.2 ± 9.2 (15)	−25.9 ± 2.8 (15)
15 Na^+^ 110 K^+^ 2.5 Ca^2+^	37	−38.6 ± 4.1 (30)	−21.2 ± 0.4 (30)	−64.1 ± 4.8 (60)	−26.7 ± 1.7 (60)
	4	−40.0 ± 3.8 (20)	−16.5 ± 2.1 (20)	−45.2 ± 0.6 (21)	−18.1 ± 1.6 (21)

Normalized distributions obtained by the Box–Cox algorithm are shown as the mean ± SEM (n = the number of impaled parasites).

## Data Availability

Data included in this article are available from the corresponding author upon request.

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
