# Peer review of "Effect of Temperature and Ionic Substitutions on the Tegumental Potentials of Protoscoleces of *Echinococcus granulosus"

_tropicalmed, 2023, doi:10.3390/tropicalmed8060303_

Round 1

Reviewer 1 Report

Dear Authors, thank you for submitting an interesting article Effect of Temperature and Ionic Substitutions on the Tegumental Potentials of Protoscoleces of Echinococcus granulosus because the transfer of information is always useful. Below you can find some comments to improve your paper.

General comments:

The paper introduces the protoscolex stage of the taeniid Echinococcus granulosus, which is responsible for generating cystic echinococcosis or hydatidosis, a globally prevalent zoonotic disease. It highlights the significance of the protoscolex's complex cellular syncytial tegument, which plays a crucial role in the regulation of ionic movements and the hydroelectrolytic balance of the parasite.

The text mentions a recent report on the existence of two electrical potentials in bovine lung protoscoleces, reflecting differences in ionic movements between the invaginated and evaginated developmental stages. This study investigated the impact of temperature and ionic substitutions on the tegumental potentials of the bovine lung protoscoleces using microelectrode impalements.

One notable finding was the temperature-dependent behavior of the transient peak potential, which exhibited an active transport component solely in the invaginated state. This suggests the presence of a temperature-sensitive mechanism involved in active transport processes. Additionally, the study observed changes in electrical potentials when high K+ depolarization, low external Ca2+, and the diuretic amiloride were introduced. These changes align with the existence of a Ca2+-sensitive, cation-selective electrodiffusional pathway on the parasite's outer surface.

The study emphasizes that the variations in electrical potential differences across the tegument provide a valuable parameter for studying ionic transport mechanisms and have the potential to identify targets for novel antiparasitic drug development.

From a critical perspective, the text provides a concise overview of the topic and highlights the significance of studying the tegumental potentials in Echinococcus granulosus. It presents findings from a recent study, providing specific experimental details such as microelectrode impalements. The results suggest potential mechanisms underlying the ionic transport and highlight the potential of the tegumental electrical potentials as targets for drug development.

Overall, while the text provides a valuable glimpse into the research on tegumental potentials in Echinococcus granulosus, it would benefit from a more thorough and critical examination of the study's methodology and its wider implications in the field.

Specific comments:

Lines 9-10 - Hydatid cyst is the larval stage of Echinococcus granulosus. Protoscolices are microscopic larvae capable of developing to adult worms in the final host intestine or to secondary hydatid cyst in the intermediate host viscera. Protoscolex is a component of the larval stage of Echinococcus species. I suggest you rephrase it so that it is not understood that the protoscolex is the larval stage.

Lines 37-38 - Please clarify to me, why you have introduced in the context of anthelmintic therapy the use of avermectins and nicotinic agonist drugs because I do not know their efficacy in the therapy of cestodoses.

The introduction is approached slightly simplistically. A few phrases are rendered that refer to Echinococcus granulosus, ionic movements and tegumental electrical potentials. The purpose of the study is not clearly stated. No reference to your results is necessary in this chapter. A state of the art should be done here.

The materials and methods chapter is too technical for a parasitologist, but it is clearly laid out, also the results chapter.

How can the potential relevance of ion channels as therapeutic targets be assessed? Are not all ion channel families known in cestodes? Are they different from one cestode species to another? If yes, which ones are known to be old and which ones are newly discovered?

Specifically, in the present study, what degree of novelty does it bring? It is not clear just from the general wording that it „provides a rapid testing approach to explore and identify new pharmacological targets”. Please be more specific.

Thank you and good luck!

Reviewer 2 Report

This manuscript describes the effect of temperature and ionic substitutions on the tegumental potentials of Echinococcus granulosus PSCs by microelectrode impalements. It is found that the transient peak potential only had a temperature-dependent consistent with an active transport component in the invaginated state. Changes in the electrical potentials by high K+ depolarization, low external Ca2+, and addition of the diuretic amiloride are in agreement with the presence of a Ca2+-sensitive, cation-selective electrodiffusional pathway in the outer surface of the parasite. It will provide a rapid testing approach to explore and identify new pharmacological targets. It is very interesting and worth publishing.

What are the criteria for the invaginated and evaginated PSCs?

Have any experiments been performed in bovine lung cells (or other cells), what are the effects on these host cells? Such as in the MS:

Effect of low temperature on host cells membrane potentials?

Effect of high external potassium concentration on host cells?

Effect of low external calcium on host cells?

Effect of amiloride on host cells?

In the references, please note the case of words! (377, 381, 423, Echinococcus granulosus)

Line15, 251, 319, 320     Echinococcus granulosus ---- E. granulosus

Line35    Please reorganize this sentence

I still have other suggestions for improvement:

1.     This manuscript needs English revision by native speaker carefully.

2.     Please follow the Authors Guidelines strictly and carefully.

1.     This manuscript needs English revision by native speaker carefully.
